# The Critical Role of 12-Methyl Group of Anthracycline Dutomycin to Its Antiproliferative Activity

**DOI:** 10.3390/molecules27103348

**Published:** 2022-05-23

**Authors:** Ruoxuan Xu, Dinghang Hu, Jinlian Lin, Jie Tang, Ruoting Zhan, Guiyou Liu, Lei Sun

**Affiliations:** 1Research Center of Chinese Herbal Resource Science and Engineering, Guangzhou University of Chinese Medicine, Guangzhou 510006, China; x13629925954@163.com (R.X.); gzucmhdh@163.com (D.H.); kimlamlin@163.com (J.L.); 20201110698@stu.gzucm.edu.cn (J.T.); ruotingzhan@vip.163.com (R.Z.); 2Key Laboratory of Chinese Medicinal Resource from Lingnan, Guangzhou University of Chinese Medicine, Ministry of Education, Guangzhou 510000, China; 3Joint Laboratory of National Engineering Research Center for the Pharmaceutics of Traditional Chinese Medicine, Guangzhou 510000, China; 4School of Life Sciences and Chemical Engineering, Jiangsu Second Normal University, Nanjing 211200, China

**Keywords:** dutomycin, anthracycline, aromatic methyl group, DNA-binding agents, molecular docking

## Abstract

Anthracycline dutomycin is a tetracyclic quinone glycoside produced by *Streptomyces minoensis* NRRL B-5482. SW91 is a C-12 demethylated dutomycin derivative, which was identified in our previous research. In vitro cytotoxicity and apoptosis assays of these two compounds were conducted to demonstrate their antiproliferation activities. The results showed that both dutomycin and SW91 block cells at the S phase, whereas dutomycin shows more significant inhibition of cell growth. Their interactions with calf thymus DNA (CT-DNA) were investigated, with dutomycin exhibiting higher binding affinity. The molecular docking demonstrated that the 12-methyl group makes dutomycin attach to the groove of DNA. These findings suggest that dutomycin has binding higher affinity to DNA and impairs DNA replication resulting in more significant antitumor activity.

## 1. Introduction

Anthracycline quinones, discovered in the 1960s, are planar molecules consisting of a rigid hydrophobic tetracycline ring, with a daunosamine sugar attached through a glycosidic bond [1]. Due to the fact of their electrophilic properties, some natural quinones interact directly with cellular nucleophiles, including soluble and protein thiol groups, and may inhibit key processes in cells [2,3]. Quinones work in three main ways. Firstly, they intercalate between base pairs to block the DNA replication and RNA transcription [4]. Secondly, anthracycline quinones stabilize the binding of nuclear topoisomerase II to DNA, resulting in protein-associated DNA strand breaks [5]. Thirdly, cytotoxic quinones exert their effects by forming redox-reactive species [6]. Anthracyclines, such as doxorubicin [7,8], have been considered as antitumor components in clinical practice for more than 50 years [9]. Due to the fact of their success in treating various types of cancer, approximately 2000 anthracycline analogs have been developed as potential anticancer agents [10], including emerrubicin, epirubicin, kamycin, pirarubicin, amrubicin, valmycin, and zorubicin [11].

Anthracyclines show good antitumor activity, though it is believed to be the main cause of chemotherapy-induced cardiotoxicity [12,13,14]. The mechanisms of the cardiotoxicity of anthracyclines are still elusive. The effects of reactive oxygen species (ROS) and topoisomerase-II (Top II) are the key factors leading to myocardial injury. ROS form toxic free radicals and reactive nitrogen species to increase nitrosative stress and mitochondrial dysfunction, which is primarily related to cardiotoxicity. Oxidative stress also leads to activation of molecular pathways leading to cardiomyocyte loss through necrosis and apoptosis. The Top II–doxorubicin–DNA ternary cleavage complex induces DNA double-stranded breaks (DSBs), leading to cardiomyocyte death [15]. Top II also acts as a regulator of various gene expression, including those involved in mitochondrial biogenesis and antioxidant function [15]. In order to enhance pharmacological activity and reduce the cardiotoxicity of anthracyclines, many studies focused on structural modifications have been undertaken [16]. For example, doxorubicin is hydroxy-substituted at the C-14 position of the A ring, which is more effective in lymphoma, sarcoma, and broad-spectrum solid tumors [11]. Due to the glycosyl group, the 4’-deoxy analog (esorubicin) shows superior pharmacological properties compared to DOX containing daunorubicin [17,18]. In addition, there are a small number of examples of modification on the D ring [18]. Idarubicin and doxorubicin differ by a methoxy substitution at the C-4 position, DOX is mainly used in solid tumors and lymphomas, and IDA is mainly used in the treatment of acute leukemia [19].

It is known that the antitumor activity of anthracyclines is mainly through the insertion of the nearly planar aromatic ring skeleton between the base pairs of DNA. The intercalation binding force mainly comes from the hydrophobicity or embedded in the minor groove region of the DNA double helix. In this way, the drug–DNA complex breaks the DNA strands and destroys the replication template [17]. In addition, anthracyclines also interfere with Top II, resulting in DNA strand breaks and the formation of ternary drug–DNA–Top II complexes, thereby affecting the structure of DNA [20,21].

In our previous work, anthracycline dutomycin was isolated from *Streptomyces minoensis* NRRL B-5482, with a tetracyclic quinone core structure and two sugar moieties, β-D-amylogenose and α-L-axenose (β-D-amicetose and α-L-axenose) (Figure 1) [22]. By knocking out the methyltransferase DutMT1, a C-12 demethylated dutomycin derivative (named SW91) was obtained (Figure 1). SW91 demonstrates higher anti-MRSA activity compared to dutomycin [23]. It has previously been reported that adding a methyl group to safinamide results in a more than 20-fold increase in potency and a nearly 1000-fold difference in selectivity [24]. In this study, we found that the 12-methyl group of dutomycin had a critical effect on its antitumor activity.

### 1.1. Purification and Identification of Dutomycin and SW91

The spectral results of dutomycin 1H-NMR (Appendix A) and SW91 1H-NMR (Appendix A) were consistent with the published data, indicating that dutomycin and SW91 were successfully purified. Detailed results are available in the Appendix A. 

### 1.2. Cytotoxicity Studies

HepG2 and Smmc-7721 were used for the in vitro assay of antiproliferative activity of dutomycin and SW91. In order to obtain the IC_50_ of the two compounds in different cell lines more accurately, we used different concentrations to study them. The dose-dependent tumor suppression is shown in the histograms of survival rate (*p* < 0.01). In HepG2 cells, the IC_50_ values of dutomycin and SW91 are 6.59 (Figure 2a) and 39.81 μM (Figure 2b), respectively. The corresponding IC50 values in Smmc-7721 cells were 7.58 (Figure 2c) and 33.20 μM (Figure 2d), respectively. The results show that both dutomycin and SW91 inhibit proliferation of tumor cells. The tumor suppression effect of dutomycin on cancer cells, Smmc-7721 or HepG2, is higher than that of SW91. Since SW91 is a C-12 demethylated dutomycin derivative, it implies that the presence of a methyl group at the C-12 position has a great impact on the antiproliferation bioactivity of dutomycin. 

### 1.3. Apoptosis Analysis

In order to demonstrate the morphology of cells, the fluorescent dye Hoechst 33342 was used for nucleus staining (Figure 3a). The cells treated with dutomycin (Figure 3b) or SW91 (Figure 3c), in the field of vision, were observed with few cell nucleus replications. While the blank control group had five cells with obvious mitosis in the field of view, which are marked with white arrows. This indicates that dutomycin and SW91 affect the nucleus replication and division. Since most anthracyclines demonstrate antitumor activity by inhibiting DNA replication, it is likely that dutomycin and its derivatives share similar mechanisms. 

Apoptotic cells were detected using FITC-labeled Annexin V (Annexin VFITC) to detect everted phosphatidylserine (PS). Propidium iodide (PI) fluoresces when it enters necrotic cells or cells lost in late apoptosis. The apoptosis analyzed using the software FlowJo7.6.5 (Appendix A). The results show that cell apoptosis is dose-dependent, and the percentage of apoptosis between the two groups of compounds is statistically significant (*p* < 0.01). As shown in Table 1, when treated with 20 μM dutomycin or SW91, the apoptosis rate of Smmc-7721 cells was 76.37 ± 5.19% and 10.45 ± 0.52%, respectively. The apoptotic rate induced by dutomycin was 6.31 times higher than that induced by SW91, which demonstrates that dutomycin has much higher antiproliferative activity than SW91. 

### 1.4. Cycle Assay

To show the effect on cell cycle caused by dutomycin or SW91, HepG2 and Smmc-7721 were synchronized in the G_0_/G_1_ period by starvation. A flow chart of cells over time after administration is shown in Figure 4. Compared to the control, there was a significant delay for the cells treated with drugs to start the G_2_ phase. The blank control starts DNA replication after 4 h incubation, whereas the drug-treated groups were detected at 12 h. 

The DNA content was monitored every 4 h after synchronization release. In the control group, the content of G_1_ gradually decreased, while the S phase increased, the highest being 61.77% at 8 h and the lowest being 29.20% at 20 h, recovering to 38.16% at 24 h. In the G_2_ period, from 0.12% at the beginning, it gradually reached 15.48% at 16 h, and recovered to 8.26% at 24 h, and the proportions in each period returned to normal (Figure 5a). The change in the cycle after adding the drug dutomycin was different: The S phase reached the highest proportion of 61.25% at 20 h and decreased by 60.62% at 24 h. This phenomenon was more obvious in the G_2_ period: the proportion in the first 16 h was basically unchanged, and there was a significant increase at 20 h, which indicates that the S-phase block phenomenon had obviously occurred (Figure 5b). The change in the period after the addition of the drug SW91 was also different. The proportion of the S period remained basically unchanged within 24 h of the detection, with only a slight increase in 12 h, and a slight decrease in the subsequent period. Compared with the control, the change was more obvious in the G_2_ period, which has been showing a slow growth state, reaching the highest proportion of 13.77% in 24 h. The results show that, compared to dutomycin, SW91 has a weaker ability to block cells in the S phase, with some cells turning to the G_2_ phase within 8 h (Figure 5c). 

In terms of cycle changes, dutomycin blocks cells in the S phase, so the proportion of the S phase is significantly increased, and the G_2_ phase is delayed. Compared to dutomycin, the ability of SW91 to block cells in the S phase was weaker, showing that the proportion of the S phase remained basically unchanged, and the increase in the G_2_ phase was faster but still lower than that of the control group. This part of the experiment showed that both act on the S phase of cells, and dutomycin is more efficient. 

### 1.5. UV–Vis Absorption Spectroscopy

UV–Vis spectrophotometry is one of the most reliable methods of detecting the interaction between compounds and DNA. The UV analysis of the solution of dutomycin and SW91 and CT-DNA in vitro showed a correlation. With the gradual addition of CT-DNA, the electronic clouds of dutomycin and SW91 interacted strongly with the ones of the DNA bases, triggering a hypochromic effect with no significant shift in peak position. This change in absorbance was mainly due to the noncovalent interactions: external contact (electrostatic bonding), hydrogen bonding, and/or groove surface bonding (primary or secondary) CT-DNA. The spectrum changed after adding these two compounds to CT-DNA, as shown in Figure 5. The absorbance of dutomycin at λ_max_ 471 nm decreased from 0.362 to 0.297 (Figure 5a), whereas the absorbance of SW91 at λ_max_ 459 nm decreased from 0.404 to 0.347 (Figure 5b). The overall decrease in the absorbance of dutomycin is more obvious. These two results are generally reflected in a small decrease in absorbance but little change in λ_max_, indicating that binding to DNA is likely through the groove [25,26]. The intrinsic binding constants (K_b_), calculated from the Wolfe–Shimmer equation (3), are presented in Table 2. The K_b_ of dutomycin was 5.86 × 10^4^/M (Figure 6a), and the K_b_ of SW91 was 4.83 × 10^4^/M (Figure 6b), indicating that the binding affinity of dutomycin to CT-DNA was higher. This process explains the interaction between the drugs and DNA, and this mode of action with DNA is likely to be the cause of their tumor suppression effect and the cell cycle arrest phenomenon. 

### 1.6. Competitive Fluorescence Measurements

In addition to UV–Vis spectrophotometry, fluorescence quenching illustrates more clearly the interaction of compounds with DNA, and competitive binding experiments were performed using ethyl sodium bromate (EB). As a sensitive fluorescent probe, EB has a planar structure and intercalates DNA to generate fluorescence. When another compound intercalates DNA, it competes with EB for the binding sites, resulting in a decrease in fluorescence intensity. The binding affinity to DNA can thus be determined by the decrease in fluorescence intensity [27,28]. With the increase in compound content, the corresponding fluorescence intensity of the fluorescence spectrum decreased. 

The fluorescence quenching experiments showed that DNA treated with either dutomycin or SW91 had an obvious decrease in fluorescence intensity. To further illustrate this result, we obtained a linear relationship through the regression equation by modifying the Stern–Volmer equation to obtain log I_0_ ⁄I = log K_b_ + n log[Q] in Figure 6. This indicates that both had competitive binding with PI, and there was an interaction between DNA. The K_sv_ quenching constant can be obtained by taking the value at the highest point of fluorescence intensity through the Stern–Volmer equation, and the K_sv_ of dutomycin was 7.67 × 10^3^/M (Figure 7a) and the K_sv_ of SW91 is 6.12 × 10^3^/M (Figure 7b), indicating that the binding affinity of dutomycin to CT-DNA-EB was higher. 

To illustrate the strength of the binding energy to DNA, the calculation results are shown in Table 2, where K_b_ is the binding constant and n is the average number of binding sites. The Kb of dutomycin was 50% higher than that of SW91. Dutomycin therefore shows a higher binding affinity to DNA (Table 3). 

### 1.7. Molecular Docking

In silico analysis using different DNA sequences as binding targets for dutomycin or SW91. Molecular docking results show that dutomycin easily inserted into the groove of DNA (PDB ID: 1BNA) (Figure 8a), mainly through hydrogen bonds. Five hydrogen bonds are predicted by molecular docking (Figure 8b). The obtain binding energy to DNA under the influence of these intermolecular forces was −7.07 kcal/mol. SW91 also intercalates into the groove of DNA (PDB ID: 1BNA) (Figure 8c), but its binding energy is lower −6.28 kcal/mol. This is mainly because there are only two hydrogen bonds between SW91 and DNA (Figure 8d). The total hydrogen bond energy of dutomycin was 77.88% lower than that of SW91 (Table 4), while the binding energy was also 11.17% lower. The DNA binding affinity of dutomycin to SW91 can also be demonstrated here.

To further explore the different binding modes of the complex to DNA, a DNA model with an intercalation gap was selected. The band results of the molecular docking software predicted that dutomycin is inserted between the duplex base pairs of DNA (PDB ID: 1Z3F) (Figure 9a). Its binding energy was −7.06 kcal/mol, and the connection between the two was mainly through hydrogen bonds. Here, we calculated the presence of two hydrogen bonds (Figure 9b). SW91 was also inserted between the double helix base pairs of DNA (PDB ID: 1Z3F) (Figure 9c) with a binding energy of −5.45 kcal/mol, and one hydrogen bond was predicted to be attached to a DG2 group (Figure 9d). The hydrogen bond energy of dutomycin to DNA (PDB ID: 1Z3F) was 36.33% lower than that of SW91, while the binding energy was lower than 22.8% (Table 4). We found that, in the process of binding to DNA (PDB ID: 4JD8), the binding mode of dutomycin to DNA was still tetra-intercalated between base pairs, and the binding energy was −8.92 kcal/mol (Figure 9e). SW91 also obtained a binding energy of −5.42 kcal/mol in the same binding mode (Figure 9g), which was 39.23% higher than dutomycin. It binds to the DNA with two hydrogen bonds (Figure 9f), whereas SW91 binds to it with three (Figure 9h). Due to the different bond energies of the hydrogen bonds, the hydrogen bond energy of dutomycin was still 18.76%, lower than that of SW91 (Table 4). Several different sequences of DNA were used as targets, and the binding energy of dutomycin was lower than that of SW91. The total hydrogen bond energy of DNA and dutomycin was also lower than SW91. This implies that the binding of DNA to dutomycin was more stable, which could explain why the anticancer activity of the compound dutomycin was higher than that of SW91.

### 1.8. Formatting of Mathematical Components

According to the equation, Stern–Volmer is used to calculate [29]:I_0_/I = 1 + K_sv_[Q],(1)
where I_0_ is the fluorescence intensity of the compound without the presence of the compound, I is the fluorescence intensity after adding the compound dutomycin and SW91, [Q] is the total concentration of the quencher, and K_sv_ is the quenching constant. The K_sv_ value is calculated from the ratio of the slope in the I_0_/I graph to the intercept of [Q]. Through the calculation of the K_sv_ value, the binding ability of the two compounds with CT-DNA can be obtained.

The binding constant (K_b_) and the average number of binding sites (n) were obtained by the modified Stern–Volmer equation [30]:log[(I_0_ − I)/I] = log K_b_ + n log[Q](2)

In this equation, I_0_ is the fluorescence intensity of the compound that does not exist, I is the fluorescence intensity after adding compound dutomycin and SW91, [Q] is the total concentration of quencher, K_b_ is the binding constant, and n is the average number of binding sites.

The value of the intrinsic binding constant (Kb) was determined using the Wolfe–Shimmer equation [30]:[DNA]/(ε_a_ − ε_f_) = [DNA]/(ε_b_ − ε_f_) + 1/Kb (ε_b_ − ε_f_)(3)
where [DNA] is the concentration of DNA; Kb is the equilibrium binding constant; ɛ_a_ is the apparent molar absorption coefficient, that is, ratio of observed absorbance (Aobs) of drug–DNA complex to drug concentration [D] (Aobs/[D]); ɛ_f_ and ɛ_b_ refer to the molar absorption coefficient of the drugs in its free and completely bound form, respectively.

## 2. Discussion

In vitro cytotoxicity test shows that both dutomycin and SW91 have good anticancer activities against human hepatoma cells. In our previous work, SW91 demonstrated a higher inhibition on MRSA, which makes it quite unexpected that human hepatoma cells are more sensitive to dutomycin. 

Fluorescence titration experiments show changes in the DNA fluorescence intensity of compounds at 298 K. The decrease in fluorescence intensity indicates that CT-DNA interacts with the compounds, and both had strong binding affinity to DNA. However, the position and shape of the fluorescence spectrum of the compounds did not change, indicating that a noncovalent bond, rather than a covalent bond, formed between the compounds and CT-DNA [31]. The difference between the two is that dutomycin still had a strong binding affinity to CT-DNA. Three DNA models were selected in this study, and the in silico assays showed that they were inserted into the grooves or the double helix base pairs of DNA. In other words, the molecular docking with these three DNAs all show low binding energies. The binding affinity was mainly affected by the van der Waals force, hydrogen bonding, and other intermolecular forces. The molecular docking experiment was consistent with the previous experimental results, indicating that the binding affinity to DNA is likely to be the reason for the difference in antitumor activity between the two compounds. 

Although there are significant differences in their antitumor activities, the experimental results show that small structural changes lead to large gaps in antitumor activity. The modification of anthracyclines on the C-ring in previous studies is mainly through metal complexation. It is well known that the modification of the C-ring can increase the stability of compounds, improve the circulation life of liposomes in vivo, and enhance the efficacy of drugs [32]. At the same time, depending on the substituent, the lipophilicity of the compound can be increased, which is beneficial for the drug to enter the cell membrane.

Both dutomycin and SW91 are hydrophobic and bind to the groove of DNA, while the 12-methyl group of dutomycin increases its hydrophilicity. Since the exposed part of DNA is hydrophilic and more likely to bind to more polar compounds, dutomycin is likely to bind to DNA more tightly than SW91. I This corresponds to the fact that molecular docking gives the lower hydrogen bond energys, of dutomycin to DNA. It is interesting that dutomycin and SW91 both showed high affinities to the GC-rich region of DNA. Furthermore, the hydrophilic dutomycin increased the solubility after passing through the cell membrane; thus, dutomycin demonstrated better pharmacological activity in cells.

According to previous work [23], dutomycin and SW91 have different activities against methicillin-resistant Staphylococcus aureus (MRSA). Here, we found that dutomycin, which is less active against S. aureus than SW91, had a better antitumor performance. Similar conditions also occurred in kibdelone A and kibdelone C. After simple structural modification, kibdelone A shows better antibacterial activity, while kibdelone C demonstrates better antitumor cell proliferation activity [33]. Another example is cannabidiol and its derivatives with the addition of one or two methyl groups. The results show that the activity against several clinically common S. aureus bacteria was significantly reduced after the addition of methyl groups [34]. Similar properties were also found in rebeccamycin and its methylated derivatives: the anti-B. cereus activity of rebeccamycin was significantly higher than its methylated derivatives. In the subsequent analysis, it is believed that the addition of methyl group reduces the antibacterial activity for various reasons. The methyl group increased the hydrophilicity, which significantly reduced the membrane permeability and greatly reduced the bioavailability in cells. On the other hand, the presence of methyl groups reduced the toxicity to topoisomerase I [34]. More research is required to fully understand the antibacterial and antitumor mechanism of anthracyclines.

This work demonstrates that dutomycin analogs appear to be highly sensitive to structural modifications on the C-ring, with small changes leading to widely different pharmacological activities. This finding provides a basis for modifying the C-ring structure of anthracyclines, which is rare in previous related studies [32]. In silico assays also provide a theoretical basis for the direction of compound structure modification. Antibacterial and antitumor experiments for a compound are also helpful in clinical applications.

## 3. Materials and Methods

The compound, dutomycin, was obtained by isolating and purifying the metabolites of the natural Streptomyces minoensis NRRL B-5482, and SW91 was obtained by isolating and purifying the metabolites of the strain in which the methyltransferase gene, Dut- MT1, was knocked out [23]. HEPES buffer (N-2-hydroxyethylpiperazine-N’-2-ethanesulfonic acid), 3-(4,5-dimethylthiazole-2)-2,5-diphenyltetrazolium bromide (MTT) ethidium bromide (EB), Hoechst 33258 (Hoechst 33342), calf thymus DNA (CT-DNA), and bovine serum albumin (BSA) were obtained from Sigma Chemical Company (New York, NY, USA). The stock solution of CT-DNA was dissolved in 0.01 M phosphate-buffered saline (PBS) buffer (Sigma, NewYork, NY, USA), pH = 7.4, with 1.8–1.9 at 260 nm and 280 nm (A260/A280), indicating that the DNA was not contaminated by proteins [35]. DNA concentration was measured at 260 nm (ε = 6600/M cm) [36].

### 3.1. Purification and Identification of Dutomycin and SW91

The two kinds of Streptomyces were cultured using YM solid medium (Yeast Malt solid medium) (Solarbio, Beijing, China), the culture condition was 28 °C, and the target product reached a higher value for 7 days, and then a double volume of ethyl acetate/methanol/ glacial acetic acid (TianjinZhiyuan, China) at 89:10:1 was added. We performed extraction, added a small amount of silica gel to the obtained dry extract after extraction to dry it, and used different concentrations of chloroform/methanol (TianjinZhiyuan, Tianjin, China) 3:1 to pick up the target components, and passed it through the chromatographic column at a flow rate of 1 mL/min. Pure product was obtained using a gradient of acetonitrile–water (Merck, Shanghai, China) (70:30, *v*/*v* to 75:25, *v*/*v* over 20 min) [23].

### 3.2. Cytotoxicity Studies

HepG2 cells (human hepatocellular carcinomas) cells were cultured in RPMI 1640 medium containing 10% fetal bovine serum, 100 U/mL penicillin, and 100 μg/mL streptomycin in a humidified atmosphere at 37 °C with 5% carbon dioxide. Smmc-7721 cells (hepatocellular cancer cell) were cultured in DMEM medium containing 10% fetal bovine serum, 100 U/mL penicillin, and 100 μg/mL streptomycin in a humidified atmosphere at 37 °C with 5% carbon dioxide.

To culture HepG2, Smmc-7721 cells (10^4^ cells/well) were treated with culture medium for 24 h, then treated with different concentrations of dutomycin and SW91 and incubated for 24 h. The set concentration was determined in advance by pre-experimentation. The control group and blank group were added with the same volume of DMSO as the dose, separately, at 37 °C in a 5% CO_2_ constant temperature incubator for 24 h; then, the supernatant was aspirated and 20 μL 3-(4,5-dimethyl-2-thiazolyl)-2,5-diphenyl-2-H-tetrazolium bromide solvent (MTT) (1 μg/mL) (Solarbio, Beijing, China) was added and continued to cultivate at 37 °C in a 5% CO_2_ constant temperature and humidification incubator so that yellow MTT and mitochondria in living cells formed. The NADP-related dehydrogenase present in the solution was reduced to insoluble blue-violet formazan, and the dead cells were not reduced. After 4 h, the supernatant was aspirated and 150 μL of DMSO was added to each well, and the formazan was dissolved by shaking on a plate shaker. Using a microplate reader (Thermo Fisher Scientific, Shanghai, China), the detection optical density of each well was detected at a wavelength of 490 nm.

The percentage inhibition was calculated by using the formula: % of Cell Viability = 100 [(Absorbance Compound Treated–Absorbance media)/(Absorbance Vehicle Treated Absorbance Media)]

We used the GraphPad Prism 8.0.1 statistical software to take the logarithm of the tumor suppression rate and made a trend line to determine the IC_50_ value [37,38].

Statistical analysis was performed with SPSS Statistics 21. The mean SEM was calculated by one-way ANOVA. Significance between groups was analyzed using the post hoc LSD test and the Shapiro–Wilk test. *p*-Values were considered significant if they were less than 0.05 and are indicated throughout using asterisks: * *p* < 0.05, ** *p* < 0.01.

### 3.3. Apoptosis Assay

We used a kit (Meilun Annexin-kFluor488/PI Apoptosis Detection Kit) (Kaiji, Jiangsu, China) to detect the occurrence of early apoptosis. First, we diluted binding buffer (10×) into 1× binding buffer working solution for later use (1 mL binding buffer (10×) needs to be added with 9 mL dd water). Cells were harvested with trypsin and washed twice with PBS without EDTA (BOSTER, Shanghai, China). We added binding buffer to the cell pellets in the drug-treated group and blank control group to 10^6^ cells/mL, pipette 100 μL (cell number 1 × 10^5^), and added 5–10 μL of PI and 5 μL of Annexin-V. These were incubated at room temperature for 15 min in the dark before adding 400 μL of binding buffer. The processed cell suspension was first filtered with a 40-mesh cell sieve and then detected by a BD-C5 flow cytometer (FACS Calibur, USA), using slow speed for detection, limiting the number of detected cells to 20,000 cells, and using FlowJo7.6.5 software for processing data [39].

### 3.4. Cell Fluorescence Experiment

The cells were aspirated and washed twice with PBS, 10% paraformaldehyde (Solarbio, Beijing, China) was added to fix the cells for 30 min at 4 °C, and 1 mL of staining solution was added to each well using Hoechst 33342 (Solarbio, Beijing, China) staining and left at room temperature for 5 min. After staining, the staining solution was removed by suction, washed twice with PBS, and photographed using a fluorescent inverted microscope [39].

### 3.5. Cell Cycle Assay

The cell cycle experiment is a traditional method used to study the effect of drugs on the cell cycle [40,41]. We used FBS-free DMEM treatment, cultured at 37 °C in a 5% CO_2_ constant temperature and humidified incubator for 24 h, so that the cancer cells were blocked in the G_0_/G_1_ period. Briefly, Smmc-7721 cells (10^6^ per well) were incubated with Medium 1640. Cells were collected with trypsin (BOSTER, Shanghai, China) and washed twice with PBS. The resuspended cells were slowly added to precooled 70% ethanol and stored at −20 °C, and at 4 °C for long-term storage. The resuspended cells were washed twice with PBS, added to PI/RNase staining solution (Solarbio, Beijing, China), and the cells were stained in the dark for 30–60 min. Cell cycle distribution was confirmed by measuring the PI fluorescence signal with a BD-C5 flowmeter (measured in 30,000 cells).

### 3.6. UV–Vis Absorption Spectroscopy

First, we configured 10 mM Tris-HCl (Zhiyuan, Tianjin, China), pH = 7.4 as the buffer solution, dissolved 1 mg/mL CT-DNA (Solarbio, Beijing, China) in 10 mM Tris-HCl, pH = 7.4 buffer solution, and detected the absorbance value of the dissolved CT-DNA solution at a wavelength of 260 nm (ε = 6600 L/M cm). According to the Beer–Lambert–Bouguer law formula:

A = εbc, where ε is the extinction coefficient, b is the optical path length, and c is the solution concentration [42].

We used 10 mM Tris-HCl, pH = 7.4 as buffer, 90 μM dutomycin, and 120 μM SW91. We added different concentrations of CT-DNA (0–34 μM), diluted to 1 mL with a 1 mL volumetric flask, and detected the UV–Vis spectrum on a Shimadzu UV–Vis spectrophotometer through a quartz cuvette with a 1.0 nm gap width. We scanned the full wavelength using a UV–Vis spectrophotometer to obtain the absorbance curve obtained by gradually increasing the CT-DNA concentration at the same drug concentration.

### 3.7. Competitive Fluorescence Measurements

The changes in the fluorescence intensity of the complex–probe–DNA solution were studied by measuring the probe addition, and the competition interaction between dutomycin and SW91 and the fluorescent probe EB (Solarbio, Beijing, China) and CT-DNA were studied [43].

First, we prepared a mixture of CT-DNA (20 μM) and EB (25 μM), dissolved it with 10 mM Tris-HCl (pH = 7.5), and added 0–200 μM dutomycin and 0–100 μM SW91; these two compounds did not have fluorescence properties. We incubated for 1 h at 25 °C for detection using a multifunctional microplate reader, where the excitation wavelength was 527 nm, and the emission wavelength was 550–750 nm.

### 3.8. Molecular Docking

Here, we used molecular docking technology to study the interaction between the complex and DNA. The 3D models of the compound dutomycin and SW91 were obtained from the website (http://www.chemspider.com/, accessed on 29 April 2006) and modified. From the protein database (http://www.rcsb.org, accessed on 24 April 2013), we obtained the typical DNA ds(CGCGAATTCGCG) 2(PDB:1BNA), ds(CGATCG) (PDB:1Z3F), and ds(ATGCAT)2 (PDB ID: 4JD8) [44,45,46]. Molecular docking uses AutoDock4.2 software to achieve molecular docking and binding energy calculations. Before docking, we removed the water molecules carried in the receptor DNA molecule and the ligand complex as well as excess ligands and added polar hydrogen for docking. Docking adopts semi-flexibility, the docking acceptor DNA molecule was determined to be rigid, and the ligand complex was sufficiently flexible to set rotatable chemical bonds. The docking pocket size was set to include all DNA models, and the rest of the parameters were default parameters. The posture with the lowest binding energy was selected as the research object for analysis using AutoDock4.2 software.

## Figures and Tables

**Figure 1 molecules-27-03348-f001:**
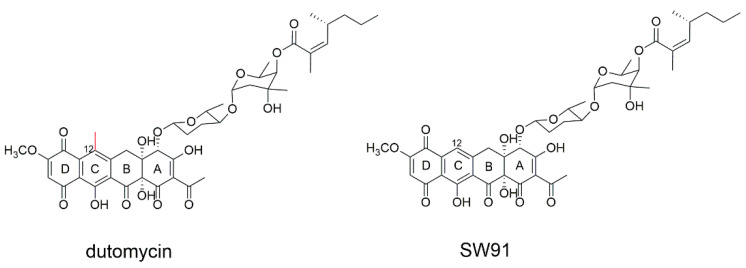
The structure of dutomycin and SW91.

**Figure 2 molecules-27-03348-f002:**
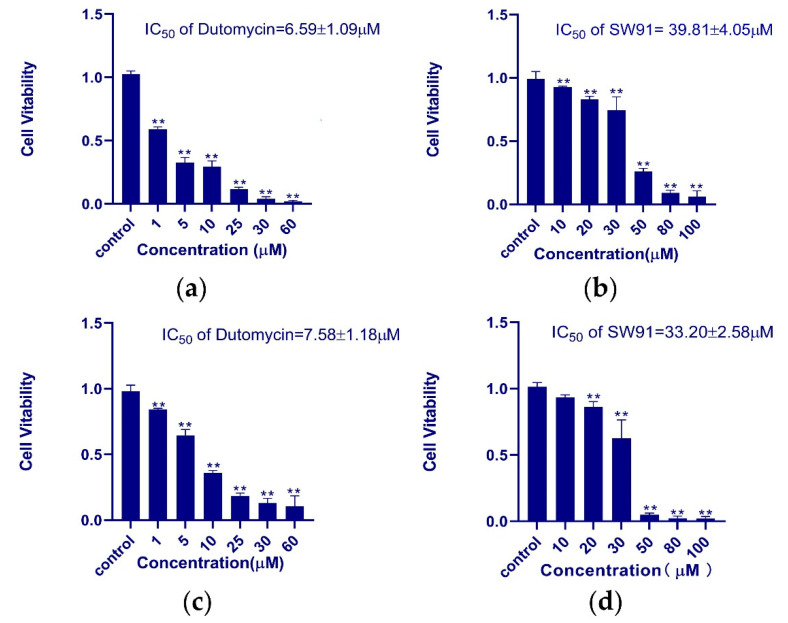
The histogram of the survival rate of dutomycin and SW91 in different cell lines: (**a**) cell survival rate in cancer cell HepG2 at different concentrations of dutomycin (μM); (**b**) cell survival rate under different concentrations of SW91 (μM) in cancer cell HepG2; (**c**) cell survival rate at different concentrations of dutomycin (μM) in cancer cell Smmc-7721; (**d**) cell survival rate at different concentrations of SW91 (μM) in cancer cell Smmc-7721. *p*-Values were considered significant compared to the control if they were less than 0.05, and they are indicated throughout using asterisks: ** *p* < 0.01.

**Figure 3 molecules-27-03348-f003:**
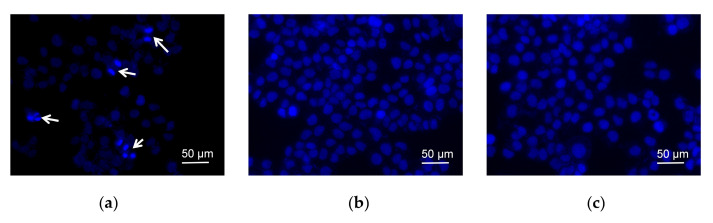
Fluorescent dye Hoechst-33342 was used for nuclear staining and changes were observed after adding different compounds; (**a**) negative control nucleus morphology and cell number; (**b**) nucleus morphology and cell number after adding SW91; (**c**) nucleus morphology and cell number after adding dutomycin.

**Figure 4 molecules-27-03348-f004:**
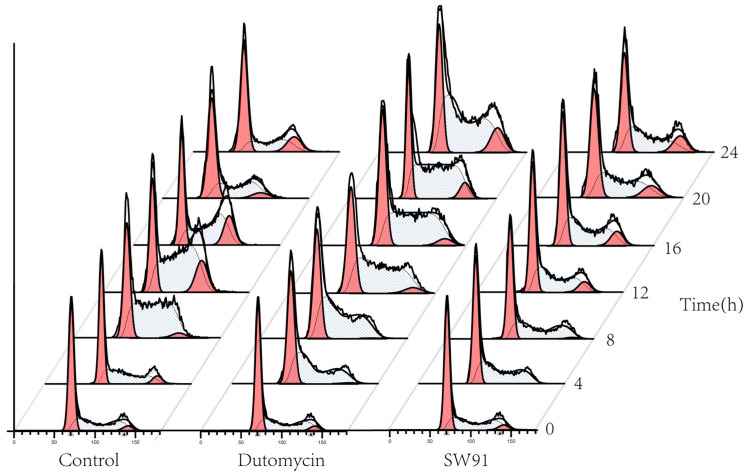
Graph of cell cycle changes for different compounds over time.

**Figure 5 molecules-27-03348-f005:**
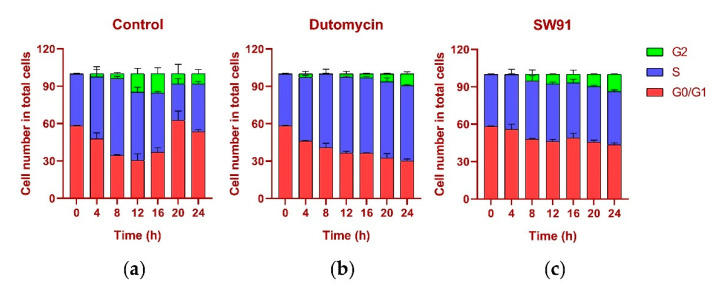
Cell cycle percentage stacking plots for different time periods 0-24 (h) to illustrate the cycles of cells treated with drugs at their IC_50_ concentrations: (**a**) plot of cell cycle percentage stacking at different time periods 0–24 (h) after addition of dutomycin; (**b**) plot of cell cycle percentage stacking at different time periods 0–24 (h) after addition of SW91; (**c**) plot of cell cycle percentage stacking at different time periods 0–24 (h) after addition of control.

**Figure 6 molecules-27-03348-f006:**
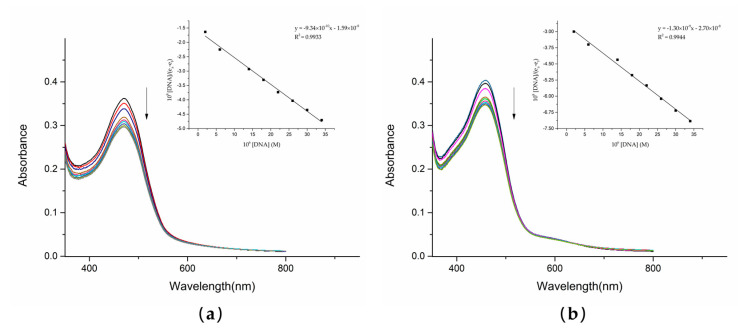
As the concentration of CT-DNA (0–34 µM) increases, the absorption spectra of dutomycin (90 µM) and SW91 (120 µM) change. The arrows’ direction indicates the increasing concentrations of DNA: (**a**) as the concentration of CT-DNA increases, the absorbance of the dutomycin system changes; (**b**) as the concentration of CT-DNA increases, the absorbance of the SW91 system changes.

**Figure 7 molecules-27-03348-f007:**
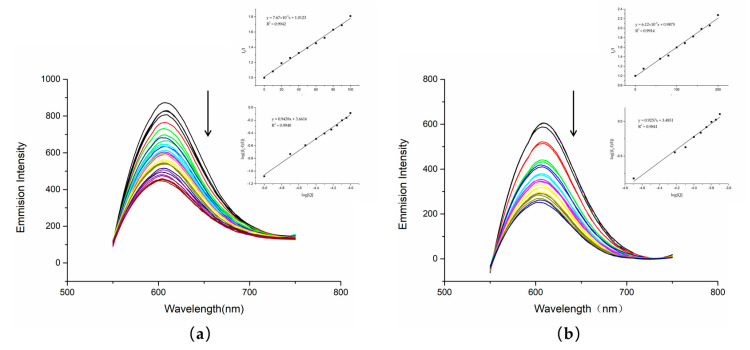
Fluorescence spectra of compounds dutomycin and SW91 in the absence or presence of CT-DNA in Tris-HCl buffer (pH 7.2): (**a**) emission spectra excited at 550 nm, c(dutomycin) = 0–100 µM, c(CT-DNA) = 20 µM, and c(EB) = 25 µM; (**b**) emission spectra excited at 550 nm, c(dutomycin) = 0–200 µM, c(CT-DNA) = 20 µM, and c(EB) = 25 µM.

**Figure 8 molecules-27-03348-f008:**
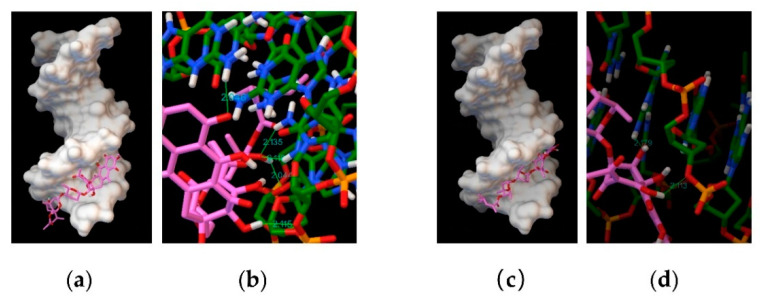
Computational docking model DNA (PDB ID 1BNA): (**a**) binding mode of dutomycin to DNA (PDB ID 1BNA); (**b**) hydrogen bond position and distance between dutomycin and DNA (PDB ID 1BNA); (**c**) binding mode of SW91 to DNA (PDB ID 1BNA); (**d**) hydrogen bond position and distance between SW91 and DNA (PDB ID 1BNA).

**Figure 9 molecules-27-03348-f009:**
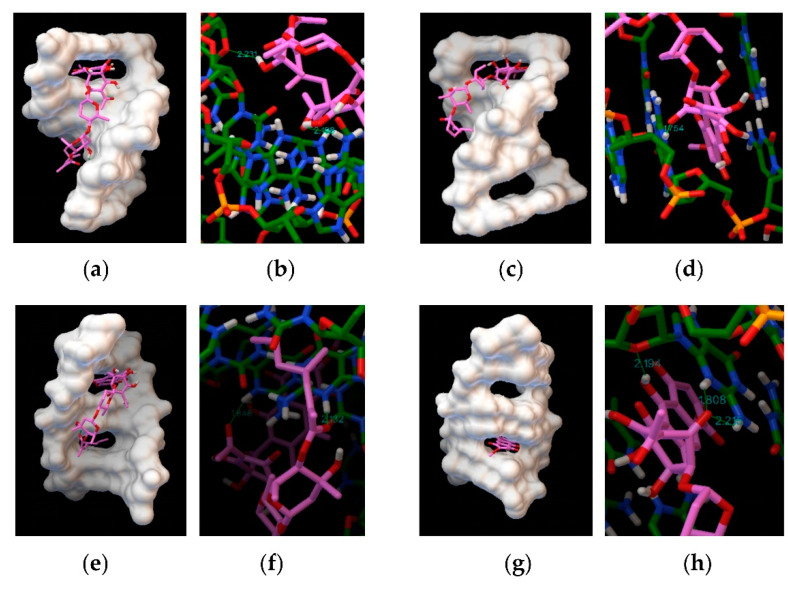
Computational docking model DNA (DNA model with gaps): (**a**) binding mode of dutomycin to DNA (PDB ID 1Z3F); (**b**) hydrogen bond position and distance between dutomycin and DNA (PDB ID 1Z3F); (**c**) binding mode of SW91 to DNA (PDB ID 1Z3F); (**d**) hydrogen bond position and distance between SW91 and DNA (PDB ID 1Z3F); (**e**) binding mode of dutomycin to DNA (PDB ID 4JD8); (**f**) hydrogen bond position and distance between dutomycin and DNA (PDB ID 4JD8); (**g**) binding mode of SW91 to DNA (PDB ID 4JD8); (**h**) hydrogen bond position and distance between SW91 and DNA (PDB ID 4JD8).

**Table 1 molecules-27-03348-t001:** Apoptosis degree of SW91 and dutomycin in cancer cell Smmc-7721.

Concentration (μM)	20	30	40	50
Dutomycin	76.37 ± 5.19	86.37 ± 0.25	93.43 ± 0.17	96.41 ± 0.47
SW91	10.45 ± 0.52	23.70 ± 2.25	45.10 ± 7.05	65.10 ± 4.12

**Table 2 molecules-27-03348-t002:** Binding constant K_b_ values of compounds dutomycin and SW91 interacting with CT-DNA.

Compound	K_b_ × 10^4^/M	R
Dutomycin	5.86	0.9933
SW91	4.83	0.9944

**Table 3 molecules-27-03348-t003:** Binding constant K_b_ values of compounds dutomycin and SW91 interacting with CT-DNA-EB.

Compound	K_sv_ × 10^3^/M	R^a^	K_b_ × 10^3^/M	n	R^b^
Dutomycin	7.67	0.9942	4.59	0.9439	0.9940
SW91	6.12	0.9914	3.06	0.9257	0.9841

**Table 4 molecules-27-03348-t004:** Hydrogen bond interactions for compounds dutomycin and SW91.

Compound	DNA	Bonds	Formed Bond Distance (Å)	Bond Energy (kcal/mol)	Total Energy (kcal/mol)
Dutomycin	1BNA	O-H…O(DA5)	2.044	−2.163	−17.327
		O-H…O(DC3)	2.115	−5.235	
		N-H…O(DG4)	2.135	−2.888	
		N-H…O(DG4)	2.118	−2.317	
		N-H…O(DG4)	2.046	−4.724	
	1Z3F	O-H…O(DC5)	2.231	−2.864	−6.849
		N-H…O(DG2)	2.198	−3.985	
	4JD8	N-H…O(DG3)	2.132	−2.71	−8.799
		N-H…O(DG3)	1.848	−6.089	
SW91	1BNA	O-H…O(DA5)	2.113	−0.586	−3.833
		N-H…O(DG4)	2.178	−3.247	
	1Z3F	N-H…O(DG2)	1.754	−6.814	−6.184
	4JD8	N-H…O(DG3)	1.808	−3.063	−7.148
		N-H…O(DG3)	2.215	−1.431	
		O-H…O(DG3)	2.193	−2.654	

## Data Availability

The authors confirm that the data supporting the findings of this study are available within the article and its Appendix A.

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
