# Peer review of "The Critical Role of 12-Methyl Group of Anthracycline Dutomycin to Its Antiproliferative Activity"

_molecules, 2022, doi:10.3390/molecules27103348_

Round 1

Reviewer 1 Report

Article “The critical role of 12-methyl group of anthracycline dutomycin 2 to its antiproliferative activity”by Ruoxuan Xu, Dinghang Hu, Jinlian lin, Jie Tang, Ruoting Zhan, Guiyou Liu and Lei Sun is devoted to the study of the use of anthracycline antibiotics as potential antitumor and antibacterial drugs.The work contains numerous experimental data demonstrating the prospects of structural modification of the C-ring in dutomycin for obtaining the most active analogues of anthracycline antibiotics.

The work is well written and corresponds to the theme of the journal. However, some minor revision are required.

  1. The text and graphs of Figure 1 do not indicate the value of error to IC50 on cell lines.
  2. 2. There is no statistical criterion for evaluating the results in the methods/under Figures.
  3. Figure 1: there is no designation ** in the caption to the figure, relative to what, what value, what statistical criterion.
  4. Why are different concentrations chosen for the same substance on different lines? And to compare two compounds? Why take concentrations with a small range, for example, as for substance SW91 (Fig. 1c)? n=?
  5. Why were these cancer cell lines chosen for research? For a more complete picture of the spectrum of antitumor activity of the compounds,itwould like to see their effect on other widely used tumor cell lines, as well as the effect on healthy cells.
  6. Figure 6: unreadable graphs
  7. In table 1, the bracket has moved out in the title of the first column.
  8. Figure 2: the scale (optical zoom) is not indicated on the photo.
  9. Line 175: ".....shown in Table 3, where Kb is the binding constant ..." these data are shown in Table 2.
  10. Figure 4 and the description for it do not indicate the concentration of the studied compounds.
  11. Have any other derivatives of dutomycin been studied, which, for example, do not have a methyl group in another position?
  12. Line 359 typo "Hcuman"

Author Response

Thank you very much for the valuable comments provided. We have addressed all the comments as shown in the revised manuscript. We have carefully edited the manuscript and figures. Molecules Research Editing Service was also hired for helping us to polish the language.

Comments from the editors and reviewers:
-Reviewer 1

Article “The critical role of 12-methyl group of anthracycline dutomycin 2 to its antiproliferative activity”by Ruoxuan Xu, Dinghang Hu, Jinlian lin, Jie Tang, Ruoting Zhan, Guiyou Liu and Lei Sun is devoted to the study of the use of anthracycline antibiotics as potential antitumor and antibacterial drugs. The work contains numerous experimental data demonstrating the prospects of structural modification of the C-ring in dutomycin for obtaining the most active analogues of anthracycline antibiotics.

The work is well written and corresponds to the theme of the journal. However, some minor revisions are required.

1.The text and graphs of Figure 1 do not indicate the value of error to IC50 on cell lines.

Response: We added the value of error to IC50 in Figure 1.

  1. There is no statistical criterion for evaluating the results in the methods/under Figures.

Response: The statistical criterion was added in the cytotoxicity studies methods in line 393.

3.Figure 1: there is no designation ** in the caption to the figure, relative to what, what value, what statistical criterion.

Response: We added the statistical criterion of “**” on legend of Figure 1.

4.Why are different concentrations chosen for the same substance on different lines? And to compare two compounds? Why take concentrations with a small range, for example, as for substance SW91 (Fig. 1c)? n=?

Response: As shown in Figure1, The IC50 of SW91 is 6-times higher than dutomycin, different concentrations were thus used for experiments. We corrected the experiment of SW91 in Figure 1b and Figure 1c, adjusted the concentration range. The concentration range adopted for different cells is obtained through pre-experiment, in order to make the cell survival rate more evenly distributed within the range.

5.Why were these cancer cell lines chosen for research? For a more complete picture of the spectrum of antitumor activity of the compounds, it would like to see their effect on other widely used tumor cell lines, as well as the effect on healthy cells.

Response: The purpose of this work is to demonstrate the difference of anti-tumor activity of dutomycin and SW91. Pre-experiments were performed on different cancer cells such as A549, MCF-7, HCT-116, HGC-27, HepG2, Smmc-7721 (seen in the figure below). It shows that the most significant different is found in the liver cancer cells Smmc-7721 and HepG2, therefore, these two cell lines were selected as major materials in experiment. Preliminary results obtained from experiments on healthy cell Base 2B show that dutomycin has lower toxicity to healthy cell Base 2B (data no show), and this result may be further explored in subsequent experiments.

6.Figure 6: unreadable graphs

Response: We revised figure 6.

7.In table 1, the bracket has moved out in the title of the first column.

Response: We adjusted table 1.

8.Figure 2: the scale (optical zoom) is not indicated on the photo.

Response: We added the scale on Figure 2.

9.Line 175: ".....shown in Table 3, where Kb is the binding constant ..." these data are shown in Table 2.

Response: We corrected this sentence.

10.Figure 4 and the description for it do not indicate the concentration of the studied compounds.

Response: We added the concentration on legend of Figure 4.

11.Have any other derivatives of dutomycin been studied, which, for example, do not have a methyl group in another position?

Response: There is only one methyl C-12 position on the core structure of dutomycin. Other similar effects of methyl are in the discussion section of the line 325-338.

12.Line 359 typo "Hcuman"

Response: We corrected this word.

Reviewer 2 Report

Section 2.5 is totally wrong. Absorption spectroscopy has been used for investigating Interaction of compounds with CT DNA by monitoring changes in absorbance at 260 nm by adding sequentially the DNA to a fixed amount of drug. This is wrong as CT DNA itself absorbs at 260 nm. If DNA is added, the increase in absorbance is due to incremental added CT DNA. It cannot be related to the interaction between compound and DNA. The correct method is to investigate at a wavelength at which the DNA does not absorb. This is normally done in visible range, 440-580 nm for most daunorubicin/doxorubicin /related derivatives . Therefore design of experiment is flawed and conclusion cannot be drawn from these experiments.   

.

Author Response

Dear Reviewers:

Thank you very much for the valuable comments provided. We have addressed all the comments as shown in the revised manuscript. We have carefully edited the manuscript and figures. Molecules Research Editing Service was also hired for helping us to polish the language.

Comments from the editors and reviewers:

Section 2.5 is totally wrong. Absorption spectroscopy has been used for investigating Interaction of compounds with CT DNA by monitoring changes in absorbance at 260 nm by adding sequentially the DNA to a fixed amount of drug. This is wrong as CT DNA itself absorbs at 260 nm. If DNA is added, the increase in absorbance is due to incremental added CT DNA. It cannot be related to the interaction between compound and DNA. The correct method is to investigate at a wavelength at which the DNA does not absorb. This is normally done in visible range, 440-580 nm for most daunorubicin/doxorubicin /related derivatives. Therefore, design of experiment is flawed and conclusion cannot be drawn from these experiments.    

Response: We redesign the experiment of DNA binding affinity assay. With the gradual addition of CT-DNA, the electronic clouds of dutomycin and SW91 interacted strongly with the ones of the DNA bases, triggering a hypochromic effect with no significant shift in peak position. This change in absorbance is mainly due to non-covalent interactions: external contact (electrostatic bonding), hydrogen bonding and/or groove surface bonding (primary or secondary) CT-DNA. The specific results are shown in Section 2.5 UV-vis Absorption Spectroscopy of the article, and the results are shown in Figure 5. 

Round 2

Reviewer 2 Report

  1. Section 2.5 - The experiments involving titrations by absorption spectroscopy have been correctly performed. Fig. 5 now shows changes at wavelength region 350-800 nm on stepwise addition of DNA to a fixed concentration of ligand – dutomycin/SW91. However, the authors should determine binding affinity using following standard equation

                      [DNA] / (εa−εf) = [DNA] / (εb−εf) + 1/ Kbb−εf)  

where [DNA] is the concentration of DNA, Kb is the equilibrium binding constant, ɛa is the apparent molar absorption coefficient, that is, ratio of observed absorbance (Aobs) of drug-DNA complex to drug concentration [D], Aobs/[D]), ɛf and ɛb refer to the molar absorption coefficient of the drugs in its free and completely bound form, respectively

The results obtained from absorption spectra should be compared with that obtained on titrations monitored by fluorescence using DNA-EB mixture (Fig. 6, Table 2). Are the binding affinities obtained by two different methods comparable??

  1. The statement “Table 2. Binding constant Kb values of compounds dutomycin and SW91 interacting with CT-DNA.” Is incorrect. It should be “Table 2. Binding constant Kb values of compounds dutomycin and SW91 interacting with CT-DNA-EB.”

  1. Section 2.6, 2nd paragraph, last line, page 4/17, line 177 : “ - - - - binding affinity of dutomycin to CT-DNA is higher”    should read as “- - - - binding affinity of dutomycin to CT-DNA-EB  is higher”

Author Response

Thank you very much for the valuable comments provided. We have addressed all the comments as shown in the revised manuscript. We have carefully edited the manuscript and figures. Molecules Research Editing Service was also hired for helping us to polish the language.

Comments from the editors and reviewers:
-Reviewer 1

1.Section 2.5 - The experiments involving titrations by absorption spectroscopy have been correctly performed. Fig. 5 now shows changes at wavelength region 350-800 nm on stepwise addition of DNA to a fixed concentration of ligand – dutomycin/SW91. However, the authors should determine binding affinity using following standard equation

                      [DNA] / (εa−εf) = [DNA] / (εb−εf) + 1/ Kb (εb−εf) 

where [DNA] is the concentration of DNA, Kb is the equilibrium binding constant, ɛa is the apparent molar absorption coefficient, that is, ratio of observed absorbance (Aobs) of drug-DNA complex to drug concentration [D], Aobs/[D]), ɛf and ɛb refer to the molar absorption coefficient of the drugs in its free and completely bound form, respectively

The results obtained from absorption spectra should be compared with that obtained on titrations monitored by fluorescence using DNA-EB mixture (Fig. 6, Table 2). Are the binding affinities obtained by two different methods comparable??

Response: The intrinsic binding constant (Kb), calculated from the Wolfe-Shimmer equation, are presented in Table 2, and the regression equation is also supplemented in Figure 5. The Kb of dutomycin is 5.86×104/M (Figure 5a), and the Kb of SW91 is 4.83×104/M (Figure 5b), indicating that the binding affinity of dutomycin to CT-DNA is higher. Compared with the fluorescence titration experiments, the Kb values of dutomycin were higher, indicating that dutomycin and DNA have higher affinity. The conclusions of the two experiments are consistent.

2.The statement “Table 2. Binding constant Kb values of compounds dutomycin and SW91 interacting with CT-DNA.” Is incorrect. It should be “Table 2. Binding constant Kb values of compounds dutomycin and SW91 interacting with CT-DNA-EB.”

Response: We corrected this statement.

3.Section 2.6, 2nd paragraph, last line, page 4/17, line 177 : “ - - - - binding affinity of dutomycin to CT-DNA is higher”    should read as “- - - - binding affinity of dutomycin to CT-DNA-EB  is higher”

Response: We corrected this sentence.